# Healthy Community Assessment Model Based on the German DGNB System

**Manshu Huang** [1], **Yinying Tao** [2,*], **Shunian Qiu** [3] **and Yiming Chang** [4]

1   School of Art, Zhejiang Shuren University, No. 8 Shuren Road, Gongshu District, Hangzhou 310009, China
2   School of Design and Fashion, Zhejiang University of Science and Technology, No. 318 Liuhe Road, Xihu District, Hangzhou 310023, China
3   School of Civil Engineering and Architecture, Zhejiang University of Science and Technology, No. 318 Liuhe Road, Xihu District, Hangzhou 310023, China
4   China Center for Information Industry Development, CCID Building, No. 66 Zizhuyuan Road, Haidian District, Beijing 100081, China
*   Correspondence: tao@zust.edu.cn

**Abstract:** As a space for daily life, the community directly affects residents' lives and has a significant impact on residents' health. Integrating the concept of health into community construction can promote comprehensive and full-cycle health protection. This study explored the potential contribution of the DGNB system to community health and well-being and collected residents' perceptions. A community assessment model was established to analyze how the community environment would affect residents' health. The results show that compared with other community evaluation systems, the DGNB system has a more balanced weight and more comprehensive content, covering many factors that influence physical health, mental health, and social health. Residents pay more attention to personal safety, lifestyle, physical environment, community service, and management, which are related to their well-being and health. The assessment model is helpful to improve the community healthy environment and residents' life quality.

**Keywords:** health; DGNB system; healthy community; local residents; healthy community assessment model

## 1. Introduction

Currently, most countries have completed urbanization, and large-scale urban construction has provided broad prospects for the life and social development of urban residents, enabling human beings to enjoy more and more advanced achievements brought about by industrialization and modernization [1]. The community environment can positively or negatively affect the health of residents. Participation within the community is positively correlated with community residents' physical and mental health [2]. Community engagement and social capital are indicators of quality of life and well-being [3]. Participation in community green spaces has positive effects on health and quality of life [4]. A community could influence resident health via the following aspects:

Community safety and community inclusion can affect community management and interaction and are determinants of healthy communities [5]. Residents' perceptions of the community, including perceived trust in neighbors, perceived community safety, and perceived community conditions, have an important impact on residents' allostatic load. A hostile community environment will tense residents psychologically, harming their physical health [6]. Green spaces around homes can buffer adverse health effects, such as tension and stress [7].

They significantly impact health and well-being, especially as people age [8]. The community environment significantly impacts older adults' well-being, with higher community accessibility, more trust among neighbors, and a cohesive community associated

with better mental health and quality of life [9]. Some scholars have suggested that the built environment will also affect the cognitive ability of the elderly [10], and community greening can reduce the probability of the elderly suffering from Alzheimer's disease [11]. Participating in meaningful activities, connecting with others, and feeling a sense of belonging to the community can help older adults maintain their mental health [12]. Even a tiny element of the built environment, such as a bench, can enhance the possibility of short periods of exercise and social interaction for older people [13].

In terms of residential buildings, the usability of housing can supplement residents' functional, social, and psychological needs, improve residents' community participation, and create good conditions for residents' healthy life [14]. There is a helpful link between indoor plants in the office and home environments and mental health, such as reduced stress, depressive symptoms, and negative mood [15]. Indoor air temperature and humidity will affect the comfort of the user's eyes and respiratory tract. The user's control of indoor ventilation is also essential to thermal comfort satisfaction [16]. A study has systematically examined the direct impact of the overall indoor environment, user social status, and lifestyle on health and studied in detail the weight of the impact of bathrooms, bedrooms, kitchens, and living rooms on health in the overall indoor environment [17].

In 2018, the World Health Organization (WHO) released the "WHO Housing and health guidelines" to improve insufficient living space, low and high indoor temperatures, indoor risk factors, and housing for people with disabilities [18]. Integrating health issues into the built environment and daily life to improve residents' quality of life and promote public health development is one of the problems that need to be solved. However, an evaluation system currently needs to evaluate communities' health from a design perspective [19]. The community design principles related to health still need to be more cohesive [20].

To fill the gap, this study establishes an evaluation system for community health based on the second-generation sustainable evaluation system DGNB and combined with residents' preferences for healthy community design principles. It aims to contribute to the health assessment of existing communities and the construction of future healthy communities and to provide a reference for healthy community design. Previous studies have demonstrated the superior performance of the DGNB system in building sustainability assessment [21–23].

This article is organized as follows:

The article first introduces the background and importance of the research topic and then explains the source and rationale of the research in Section one and Section two. Section three surveys urban residents' preferences and opinions on healthy community design principles, screens existing evaluation indicators, and classifies various factors that affect community health. The fourth part establishes a community health evaluation system to comprehensively evaluate the impact of a community design on residents' health. The last section includes the conclusion, limitations, and future research.

## 2. The DGNB System

DGNB is the abbreviation of Deutsches Guetsiegel Nachhaltiges Bauen. This system was developed jointly by the German Ministry of Transport, Construction, and Urban Planning (BMVBS) and the German Sustainable Building Council.

The DGNB system is the leader in providing construction project certification in Germany; the system accounts for more than 60% of the entire German commercial real estate market, and more than 80% of new buildings in Germany are certified by DGNB [24]. The DGNB Association currently has approximately 1200 members. It promotes an understanding of sustainable development issues in the construction industry by creating an online platform that provides access to expert knowledge on sustainable construction, construction product listings, and all other relevant information. The DGNB system can adapt to climate, structural, legal, and cultural changes in other countries, enabling certification on a global scale while maintaining high-quality standards. All the above make DGNB the most significant sustainable building network in Europe, and it has a strong influence

worldwide [25]. The DGNB system uses performance indices to grade projects. The overall project performance index is derived by calculating the weighted scores of the six topics. The certificates include Platinum, Gold, Silver, and Bronze.

### 2.1. Origin of DGNB

From the perspective of construction development, Germany has always been the first to put forward requirements for construction standards. At the beginning of 1952, some German institutions began to propose relevant minimum standard requirements for building insulation. Furthermore, Germany officially promulgated and implemented the first energy efficiency requirements for insulation in 1978 [26]. Since then, the relevant requirements have become more stringent. Beginning in 1990, German architects began to develop high-efficiency passive houses. The Passive House Institute (PHI) in Germany is in a leading position in the research and development of passive house design, components, and energy-saving quality assurance [27]. By 2002, the Energy Conservation Ordinance (EnEV) came into effect [28]. In 2008, based on the previous energy-saving building regulations, Germany formally proposed the second generation of sustainable evaluation system—DGNB.

Compared with other green building evaluation standards, DGNB includes "green" factors and covers economic, social, technical, construction, and site factors. The sociocultural and functional quality sections also include aspects related to the healthy design of indoor and outdoor environments.

### 2.2. Horizontal Comparison

In the 1990s, the British government agency BRE (Building Research Establishment) formulated the first building green evaluation standard, BREEAM (Building Research Establishment Environmental Assessment Method), and the US USGBC (U.S. Green Building Council) launched the LEED (Leadership in Energy and Environmental Design) evaluation system. Healthy communities and buildings have also started the standardization process, such as the WELL Building Standard (WELL) in the United States. The Architectural Society of China released the "Assessment Standard for Healthy Building T/ASC02-2016" in 2017.

Therefore, the building standards selected for comparative analysis were DGNB System New Construction Buildings (Version 2020 International), LEED v4.1 Residential BD+C Multifamily Homes 2020, BREEAM International New Construction Version 6.0, The WELL Building Standard™ version 2 Q4 2021, and T/ASC02-2016 "Assessment standard for healthy building". The community standards selected for comparison were DGNB System Districts (Version 2020), LEED v4 for Neighborhood Development, BREEAM Communities, and WELL Community Standard Q2 2021.

The main reason for choosing the above evaluation criteria is that the above criteria are directly related to healthy design. Although they are called green building evaluation standards, they include many contents and terms of healthy design. Secondly, the evaluation criteria mentioned above have a high popularity and wide market acceptance. In addition, all can apply to community and residential designs.

LEED v4.1 Residential BD+C Multifamily Homes 2020 is an evaluation system dedicated to residential buildings in the LEED family, applicable to single-family and multifamily houses. It can be used for LEED Online registration in all countries and regions [29]. LEED v4 for Neighborhood Development (LEED ND) was engineered to inspire and help create better, more sustainable, and well-connected neighborhoods. It looks beyond the scale of buildings to consider entire communities [30].

Launched in 1990, BREEAM is the world's first leading building environmental sustainability assessment and certification scheme. So far, it has been used to certify more than 590,000 building assessments throughout the building life cycle and has been used in more than 85 countries and regions. BREEAM aims to mitigate the environmental impact of buildings during their life cycle, enable buildings to adapt to local environments, provide buildings with a credible sustainable certification label, and stimulate the demand for

sustainable buildings [31]. BREEAM Communities International can be used to improve, measure, and certify the sustainability of community plans, including new communities and regeneration projects. BREEAM Communities is being applied internationally, and certified assessments of developments can be found across Europe, the Middle East, Africa, and Asia [32].

WELL is a standard that focuses on the health and well-being of building users. Its purpose is to improve the design strategy of human health and comfort and to reshape buildings that are equally beneficial to the Earth and humans. The latest version, WELL version 2, is a globally applicable rating system that can be adapted to the requirements of any setting or organization seeking to improve human health and promote the well-being of all. Multifamily buildings can also be certified with WELL version 2, which makes up for the limitation that version 1 only applies to commercial and institutional office buildings [33]. The WELL Community Standard focuses on the entire public space in which people spend their daily lives. WELL Communities are designed to support health and well-being in all aspects and areas of community life. The vision for the WELL community is to be inclusive, comprehensive, and resilient with a strong sense of community identity that fosters high levels of social interaction and engagement. The resources in the WELL community—natural, human, and technological—are used efficiently, equitably, and responsibly to meet the community's current and future needs and priorities [34].

The Architectural Society of China Standard T/ASC02-2016 "Assessment standard for healthy building" was formulated by the China Academy of Building Research, the China Urban Science Research Association, and the China Architectural Design Institute Co., Ltd. with relevant units. It came into effect on 6 January 2017. The standard follows the principle of multidisciplinary integration. It establishes evaluation criteria covering three aspects of physiology, psychology, and society as the first-level evaluation criteria: air, water, comfort, fitness, humanities, services, and promotion and innovation. The index classification is similar to WELL. The standard considers the building health performance evaluation in two stages of design and operation, is suitable for residential buildings and public buildings, and sets the evaluation index weights accordingly. The health performance grades of healthy buildings are divided into three grades according to the total score [35].

### 2.2.1. Evaluation System Weight Comparison

The topics of each evaluation system have different weight assignments. The sustainability concept of the DGNB system is broad in scope and comprehensively covers almost all fundamental aspects of sustainable architecture. These topics include ecology, economy, sociocultural and functional criteria, technology, process, and site. Economic and sociocultural criteria have the same weight in DGNB as ecological criteria, making the DGNB system the only system that places equal emphasis on economic aspects and ecological criteria of sustainable buildings. Criteria, such as technology, operation management, and site, have interdisciplinary functions and different weights, which makes the scope of the DGNB evaluation extend to the entire life cycle of buildings.

LEED adopts a graded scoring system, which consists of basic items that must meet the standards and scoring items, with a total score of 110 points. The weight of LEED combines quantitative and qualitative criteria and pays more attention to the green ecological aspects of the project's geographical location, energy and atmosphere, and indoor physical environment. Innovation is included in the additional part, and its proportion is relatively small. BREEAM has subdivided natural resources and has detailed regulations on water resources, materials, waste, land use, and pollution. Compared with other sustainable building evaluation systems, BREEAM raises building management and users' health and well-being to the same important position as energy and resources, pays more attention to the sustainable development of building users, and has a 10% additional innovation point. Due to WELL's unique score calculation method, all its scoring items have the same weight to ensure that the project can achieve the health requirements of each scoring item in a balanced manner, and they also have 10% additional innovation points. T/ASC02-2016

"Assessment standard for healthy building" focuses on building users' environmental and behavioral health. It does not include the evaluation of building management services in the design evaluation and also includes at most ten additional points for improvement and innovation.

Figure 1 shows the weight, content, and proportion of the highest level in the above evaluation system (excluding additional points):

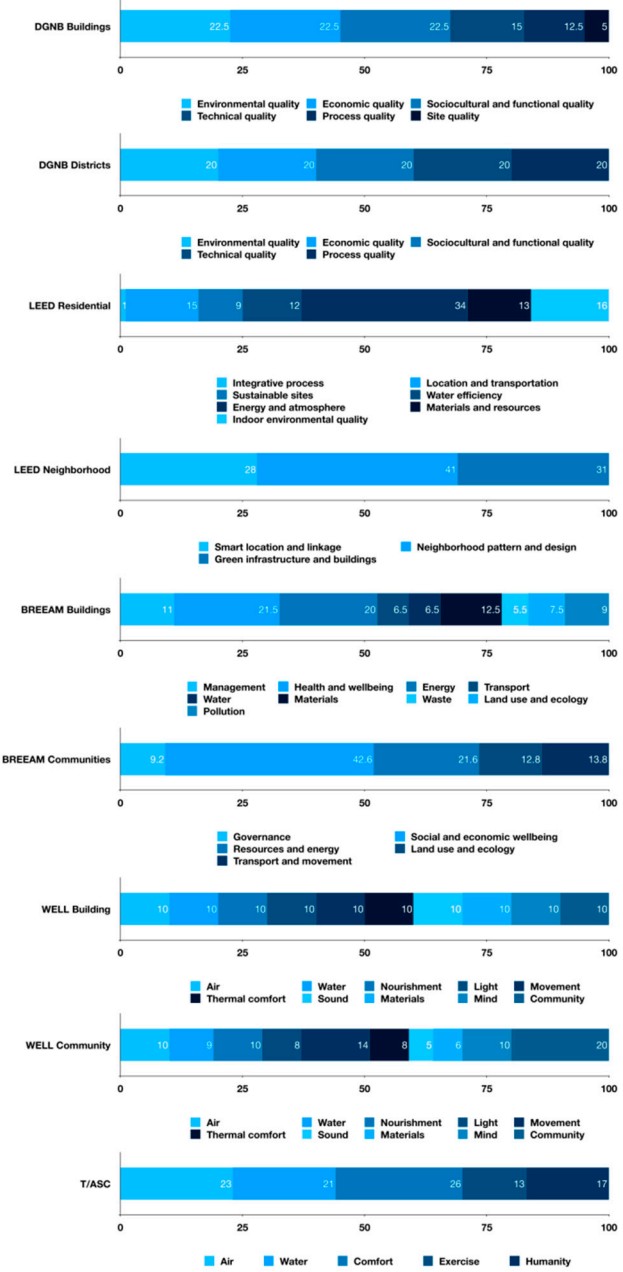

**Figure 1.** Criteria of content and weight of each evaluation system (adapted with permission from Refs. [24–31]).

The criteria of each evaluation system are summarized into three dimensions: environment, economy, and society as shown in Figure 2. The criteria of DGNB Buildings, DGNB Districts, LEED Neighborhood, and BREEAM Communities are relatively balanced in the three aspects of environment, economy, and society. LEED Residential, BREEAM Buildings, WELL, and "Assessment standard for healthy building" all have the lowest or no economic dimension.

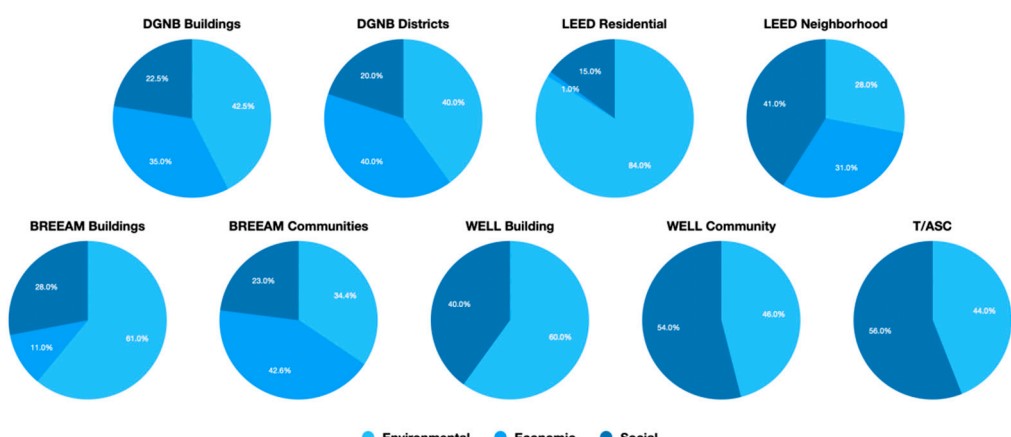

**Figure 2.** Comparison among DGNB, LEED, BREEAM, WELL, and "Assessment standard for healthy building" (adapted with permission from Refs. [24–31]).

### 2.2.2. Evaluation System Content Comparison

Table 1 shows the distribution of criterion categories for each system. The same or similar criteria are combined to compare and display the standard and missing categories among the various systems. The content comes from each system, and the similar categories were reorganized statistically by the authors. Due to the large number of specific criteria, for some systems, the absence of some categories does not mean that the system lacks the evaluation content of this section. It may also be that the relevant content is attributed to another section.

**Table 1.** Evaluation content comparison (reproduced from [24–31]).

|  | DGNB | LEED | BREEAM | WELL | T/ASC |
|---|---|---|---|---|---|
| Building Life Cycle | √ |  | √ |  |  |
| Design Process Integration |  | √ | √ |  |  |
| Local Environment | √ | √ | √ |  |  |
| Water Quality | √ | √ | √ | √ | √ |
| Conserve Water | √ |  | √ |  |  |
| Energy and Resources | √ | √ | √ |  |  |
| Land Use | √ | √ | √ |  |  |
| Ecosystem | √ | √ | √ |  |  |
| Flexibility and Adaptability | √ |  |  |  |  |
| Business Potential | √ |  |  |  |  |
| Development Maturity |  | √ |  |  |  |
| Economic Cost | √ |  | √ |  |  |
| Indoor Physical Environment | √ | √ | √ | √ | √ |
| Air Quality | √ | √ | √ | √ | √ |
| Noise Control | √ | √ | √ | √ | √ |
| Thermal Comfort | √ | √ | √ | √ | √ |
| Visual Comfort | √ | √ | √ | √ |  |
| User Control | √ |  |  |  |  |
| Public Space Quality | √ |  |  |  | √ |
| Safety | √ |  | √ |  |  |
| Accessible Design | √ | √ | √ |  | √ |
| Ergonomic Design |  |  |  |  | √ |
| Building Envelope | √ |  |  |  |  |
| Technical Equipment | √ |  |  |  | √ |
| Materials | √ | √ | √ |  | √ |
| Pollution Control | √ | √ | √ | √ | √ |

**Table 1.** *Cont.*

| | DGNB | LEED | BREEAM | WELL | T/ASC |
|---|:---:|:---:|:---:|:---:|:---:|
| Material Recycling | √ | √ | √ | | |
| Waste Management | | √ | √ | | |
| Transport Infrastructure | √ | √ | √ | | |
| Quality Transportation | √ | √ | √ | | |
| Nonmotorized Facilities | | √ | √ | | |
| Walkable Street | | √ | | | |
| Daily Life Facilities | √ | √ | √ | | |
| Design Procedure | | | | | |
| Operation Management | √ | | √ | | √ |
| Systematic Commissioning | √ | | √ | | |
| Construction Site | | √ | | | |
| Construction Quality | √ | √ | √ | | |
| User Communication | √ | | | | √ |
| Nutrition | | √ | | √ | |
| Fitness | √ | | | √ | √ |
| Mental Health | | | | √ | √ |
| Innovation | | | | | |
| Regional Priority | | | | | |

The table shows that each system has its classification of evaluation criteria, but each index category also has similar content. Although the names of the various systems are different and the criteria are not the same, they are similar in terms of the project's green, sustainable, and healthy aspects. This table also shows that some evaluation systems have unique evaluation content, such as DGNB's investigation of a project's commercial potential, flexibility and adaptability, and user control, which can reflect its superiority over other systems.

In terms of evaluation content, in most cases, "green" and "healthy" are common themes of the above evaluation systems. In LEED and BREEAM, the use of energy, the quality of the indoor physical environment, and whether the building materials are harmful occupy a relatively high proportion of the score. These two pay more attention to the impact of buildings on the natural environment and human physical health. However, they do not explicitly involve building users' comfort and mental health. WELL and T/ASC02-2021 "Assessment standard for healthy building" start from building users, and all scoring standards serve users without considering the protection of the environment. They are relatively pure health evaluation standards. As a second-generation building evaluation system, DGNB includes traditional green building evaluation items, such as environmental protection, building materials, construction process, and construction sites, and also evaluates social culture, economic costs, and event venues. Moreover, DGNB pays attention to the later sustainable operation management and interactive communication with users. Although the number of DGNB scoring items is not the largest, the weight of each aspect is relatively average, and the evaluation content can be considered the most comprehensive.

### 3. Materials and Methods
*3.1. Survey*

In order to clarify the criteria related to the healthy community in the DGNB system and to establish a healthy community evaluation system based on the DGNB system, a questionnaire survey was designed to collect statistics on residents' understanding of various criteria in DGNB. Respondents were asked to rate the impact of each factor in the questionnaire on their quality of life, physical health, and mental health in daily life: No impact at all = 1, No impact = 2, General = 3, Influential = 4, and Very influential = 5. A score of 1 means that the factor is entirely irrelevant to health, and a score of 5 means that the factor is very much related to health. The higher the score, the higher the residents



think this aspect affects their health. The final score was obtained by calculating the average scores for each topic [36].

The questionnaire questions were based on the DGNB system. In order to avoid repeated questions, similar criteria of DGNB Buildings and DGNB Districts were merged, including all criteria in the DGNB system. In order to allow respondents to understand better the criteria represented by the topics, the questionnaire used language that is as close to daily life as possible and easy to understand.

### 3.2. Variables

The questionnaire survey was conducted in China through a professional open online questionnaire tool. The questionnaire was collected from September 2020 to December 2021. A total of 500 questionnaires were distributed, and 487 questionnaires were returned, including 439 valid questionnaires. The survey was designed to be representative of the population of urban residents in China's Yangtze River Delta. So the respondents were basically from Shanghai City, Jiangsu Province, and Zhejiang Province. An overview of the variables used in the analyses is presented in Table 2.

**Table 2.** Basic demographic information of the respondents.

| | | |
|---|---|---|
| Gender | Male | 215 (49%) |
| | Female | 224 (51%) |
| Age | 20–29 | 202 (46%) |
| | 30–39 | 92 (21%) |
| | 40–49 | 66 (15%) |
| | 50–59 | 53 (12%) |
| | Over 60 | 26 (6%) |
| Occupation | Student | 189 (43%) |
| | Employed | 198 (45%) |
| | Unemployed | 52 (12%) |

### 3.3. Statistical Analysis

All the data from the questionnaire survey were saved on data extraction sheets, digitized, and stored in SPSS data files using IBM SPSS Statics 24. The data were analyzed by descriptive statistics using proportions of demographic characteristics and the scoring data of the residents. We used nonparametric tests and Pearson's chi-squared test to identify significant differences in scoring data among age groups and occupational statuses with a statistical significance threshold of $p < 0.05$ [37].

The health-related criteria in DGNB were further screened through residents' scoring. The scores of all topics, the proportion of each score, and the average scores are shown in Table 3.

**Table 3.** Score statistics.

| Category | Topic | No Impact at All (%) | No Impact (%) | General (%) | Influential (%) | Very Influential (%) | M (SD) |
|---|---|---|---|---|---|---|---|
| Resources and Environment | Community consumption of nonrenewable resources | 299(68) | 66(15) | 40(9) | 26(6) | 9(2) | 1.59 (1.01) |
| | Community use of hazardous materials and products | 4(1) | 18(4) | 44(10) | 105(24) | 268(61) | 4.40 (0.90) |
| | Utilization of resources and energy in the community | 88(20) | 202(46) | 61(14) | 53(12) | 35(8) | 2.42 (1.17) |
| | How resources are extracted and processed in communities | 228(52) | 101(23) | 75(17) | 31(7) | 4(1) | 1.82 (1.01) |
| | Recovery and reuse of resources in the community | 171(39) | 127(29) | 92(21) | 40(9) | 9(2) | 2.06 (1.07) |
| | Recycling and reuse of wastewater in the community | 9(2) | 18(4) | 149(34) | 233(53) | 31(7) | 3.59 (0.76) |
| | Community local biodiversity | 26(6) | 57(13) | 114(26) | 215(49) | 26(6) | 3.36 (0.99) |

**Table 3.** *Cont.*

| Category | Topic | No Impact at All (%) | No Impact (%) | General (%) | Influential (%) | Very Influential (%) | M (SD) |
|---|---|---|---|---|---|---|---|
| Economy | Community construction and postmaintenance costs | 18(4) | 44(10) | 75(17) | 246(56) | 57(13) | 3.64 (0.97) |
| | Community land use | 158(36) | 145(33) | 48(11) | 53(12) | 35(8) | 2.23 (1.27) |
| | The commercial value of the community | 9(2) | 224(51) | 97(22) | 83(19) | 26(6) | 2.76 (0.98) |
| Use | Flexibility and freedom of interior space functions | 0(0) | 13(3) | 40(9) | 97(22) | 290(66) | 4.51 (0.78) |
| | Indoor and outdoor temperatures | 4(1) | 44(10) | 79(18) | 215(49) | 97(22) | 3.81 (0.93) |
| | Indoor air quality | 0(0) | 0(0) | 31(7) | 101(23) | 307(70) | 4.63 (0.61) |
| | Indoor natural and artificial lighting | 4(1) | 35(8) | 48(11) | 233(53) | 119(27) | 3.97 (0.89) |
| | Indoor view of the outdoors | 18(4) | 66(15) | 70(16) | 193(44) | 92(21) | 3.63 (1.09) |
| | Residents' adjustable control of the indoor environment | 0(0) | 0(0) | 26(6) | 110(25) | 303(69) | 4.63 (0.59) |
| | The function and quality of community indoor public space | 26(6) | 35(8) | 48(11) | 176(40) | 154(35) | 3.90 (1.15) |
| | The function and quality of community outdoor open spaces | 22(5) | 48(11) | 114(26) | 136(31) | 119(27) | 3.64 (1.14) |
| | Community safety and security | 0(0) | 13(3) | 40(9) | 119(27) | 268(61) | 4.46 (0.78) |
| | Accessible facilities in the community (including ramps, elevators) | 4(1) | 35(8) | 202(46) | 127(29) | 70(16) | 3.51 (0.89) |
| Technology | Quality of residential exterior walls | 22(5) | 61(14) | 198(45) | 61(14) | 97(22) | 3.34 (1.12) |
| | Energy-saving technology adopted in residence | 4(1) | 290(66) | 75(17) | 57(13) | 13(3) | 2.51 (0.84) |
| Community | Transport infrastructure (including roads, public transport, and parking lots) | 0(0) | 9(2) | 35(8) | 268(61) | 127(29) | 4.17 (0.65) |
| | Green transportation facilities (including motor/nonmotor vehicles charging piles, and shared bicycles) | 26(6) | 31(7) | 48(11) | 171(39) | 162(37) | 3.94 (1.14) |
| | Convenience of daily life (including shopping, medical care, education, entertainment, and other public services) | 0(0) | 4(1) | 40(9) | 136(31) | 259(59) | 4.48 (0.70) |
| | Motor vehicle travel environment | 18(4) | 26(6) | 193(44) | 145(33) | 57(13) | 3.45 (0.93) |
| | Pedestrian and nonmotorized travel environment | 9(2) | 53(12) | 61(14) | 206(47) | 110(25) | 3.81 (1.01) |
| Construction | Community planning and design process | 79(18) | 294(67) | 57(13) | 9(2) | 0(0) | 1.99 (0.63) |
| | Community construction process | 140(32) | 206(47) | 83(19) | 9(2) | 0(0) | 1.91 (0.76) |
| | Residential construction quality | 0(0) | 0(0) | 13(3) | 184(42) | 241(55) | 4.52 (0.56) |
| | System check before handover | 18(4) | 26(6) | 66(15) | 272(62) | 57(13) | 3.74 (0.90) |
| Management | Information communication between residents and managers | 18(4) | 57(13) | 88(20) | 158(36) | 119(27) | 3.69 (1.12) |
| | Community culture, activities, and relationships | 18(4) | 40(9) | 145(33) | 167(38) | 70(16) | 3.53 (1.00) |
| | Residents participate in community governance | 22(5) | 97(22) | 259(59) | 53(12) | 9(2) | 2.84 (0.77) |
| | Community impact on local areas | 40(9) | 61(14) | 255(58) | 70(16) | 13(3) | 2.90 (0.88) |

Note: M = Mean; SD = Standard Deviation.

The questionnaire survey results show that residents have a relatively clear view of the relationship between the evaluation criteria in the DGNB system and health.

The results of the questionnaire survey show that topics with scores in the 4–5 range indicate that residents generally pay more attention to aspects that will directly affect their health. For example, hazardous materials will release harmful substances that directly affect the human body and endanger health. Community safety, transportation, surrounding shopping, medical care, and other aspects directly affect residents' basic living needs and convenience, so the scores are also high. Although the scores of topics between 3 and 3.5 are not in the highest range, they also reflect the more profound and richer needs and tendencies of most residents for health. For example, indoor and outdoor temperatures, daylighting, and lighting represent the leading physical environment quality indoors and significantly influence the user's physical comfort and visual comfort. Outdoor scenery, quality of public space, quality of outdoor open space, community cultural activities, and other aspects reflect the residents' pursuit of quality of life beyond basic living needs. Accessible facilities in the community reflect residents' concern and love for the elderly and the disabled. The review before the community handover and the information communication of the property show that most residents are paying more and more attention to the management

of the community. Most topics with an average score of less than two focus on the aspects of resources, energy, adopted technologies, and the precommunity construction process, which are less relevant to the daily life of residents, so the scores are lower.

## 4. Assessment Model Development Based on DGNB

In 1948, the World Health Organization defined health as "A state of complete physical, mental and social well-being and not merely the absence of disease or infirmity" [38]. The impact of the environment on health can be divided into three dimensions: physical health, mental health, and social health [39]. The measured variables of physical health mainly include the degree of bodily pain, the ability to complete work, and whether daily activities are affected by physical health. The measured variables of mental health mainly include feeling cheerful, calm, relaxed, active, vigorous, and full of interest in life. Social health mainly includes community residents being able to help each other, trust, get along with each other, and solve problems together.

In the results of the Likert scale, it can be considered that topics with a score higher than 3.5 are strongly correlated with health [40]. Referring to the respondents' scores on the health-related criteria in the DGNB system in the previous questionnaire survey, select topics with scores higher than 3.5, and combine the three dimensions of health to establish a healthy community assessment model. All twenty-one topics with scores higher than 3.5 were organized into fourteen criteria and five thematic dimensions, namely Ensure the personal safety of residents, Flexibly adapt to different lives, Optimize indoor environmental quality, Promote a healthy lifestyle, and Build operation management system. We selected the dimensions based on the DGNB system (Figure 3 and Table 4).

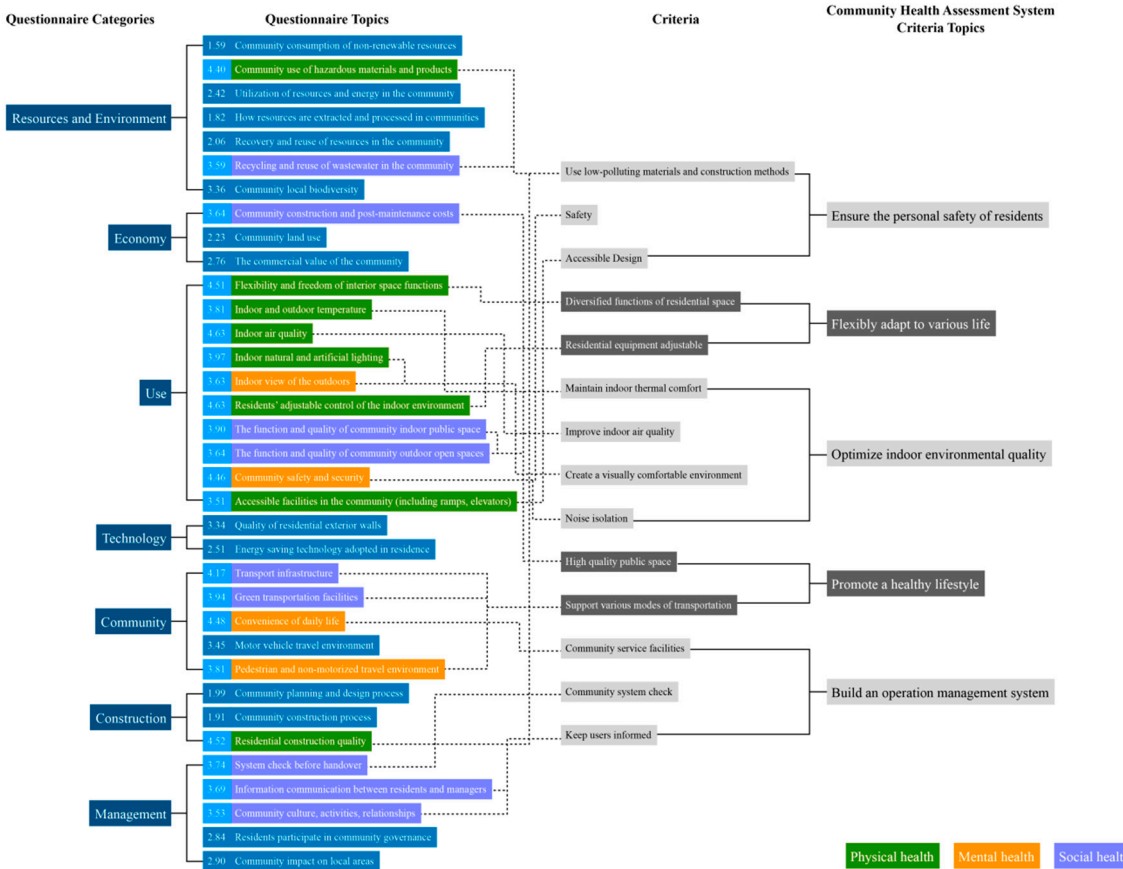

**Figure 3.** A conceptual model of health's three dimensions, the questionnaire results, and the health assessment system.

**Table 4.** Community Health Assessment System.

| Criteria Topic | Criterion |
| --- | --- |
| Ensure the personal safety of residents | Use low-polluting materials and construction methods |
| | Safety |
| | Accessible design |
| Flexibly adapt to various life | Diversified functions of residential space |
| | Residential equipment adjustable |
| | Maintain indoor thermal comfort |
| Optimize indoor environmental quality | Improve indoor air quality |
| | Create a visually comfortable environment |
| | Noise isolation |
| Promote a healthy lifestyle | High-quality public space |
| | Support various modes of transportation |
| | Community service facilities |
| Build an operation management system | Community system check |
| | Keep users informed |

### 4.1. Ensure the Personal Safety of Residents

4.1.1. Use Low-Polluting Materials and Construction Methods

Some construction products and preparations are hazardous to the soil, air, ground and surface water, people, plants, and animals [41]. Hazardous substances adversely affect the environment and people throughout their life cycle from manufacturing, processing, use, and final disposal (demolition, recycling, and landfill). Therefore, in architectural design, outdoor environment design, and interior design, it is necessary to reduce, avoid, or replace all substances or materials that may have adverse effects on humans, animals, and plants and reduce damage to residents' health.

Among them, the critical points of concern are the basic structure, exterior wall structure, interior wall structure, floor and ceiling structure, roof structure, and underground garage of the house. Regarding residential components, attention should be paid to whether the coatings, doors and windows, floors, wall coverings, sealants, wood, and other materials on the construction site contain harmful substances.

4.1.2. Safety

Community safety is positively associated with mental health [42]. The community should take building measures that increase people's sense of safety and prevent dangerous situations in and around homes.

Firstly, improving the subjective perception of residents' security and defense against attacks is necessary by improving visibility and maintaining adequate lighting in entry areas, main thoroughfares, garages, and parking lots. Secondly, components to improve privacy, such as blinds, can be added to the exterior windows of houses. In addition, some technical security facilities can also be adopted, such as smart community security systems. The intelligent community security system monitors the community in real time through the alarm system, face recognition system, and video surveillance system around the residence. The system can also report abnormal events inside the house for emergency treatment. Through IoT technology and intelligent technology, the security inside and outside the house is further strengthened.

4.1.3. Accessible Design

Due to the resident population's diversity, the house's overall environment should be equally accessible and usable by all [43]. Regardless of their circumstances, their use should not be restricted.

Accessible design is usually aimed at the following groups: the disabled, the elderly, children, and other physically or mentally disabled people, including those with long-term impairments in body, language, hearing, spirit, intelligence, or many other aspects. In

addition, accessible design can also help healthy people (such as those who carry heavy objects or push strollers).

The accessible design of the community includes barrier-free entrances, barrier-free passages, and barrier-free elevators [36]. Barrier-free toilets should be set up in the public areas of residential buildings, and unique parking spaces and circulation areas should be set up for the disabled. According to the actual situation of the residence, it is required to set up barrier-free housing or barrier-free floors, and it is recommended to set this up on the ground floor of the building. Barrier-free housing should consider the ability and habits of particular groups of people and make designs regarding room function, area, size, and furniture size. For people with language, visual, and hearing impairments, the "multisensory principle" should also be adopted: residents can use at least two or more senses of vision, hearing, and touch to ensure the regular progress of daily activities and spatial cognition [44]. For people who have doubts about using residential facilities and cannot understand them, more readable and intelligent methods should be used to convey information and reduce the difficulty of using residential facilities. It can also provide artificial assistance services for particular groups of people, and convenient one-key calling or emergency equipment can be set up in public areas and in the residences of residential buildings to prevent emergencies.

If the principles of accessible buildings have been incorporated in the planning of the residence, these designs can avoid later reconstruction of the building. The accessible design makes buildings more attractive to all user groups. As the degree of aging deepens and the population structure changes, differences among people should be tolerated and encouraged to open more possibilities for healthy housing.

### 4.2. Flexibly Adapt to Different Lives

4.2.1. Diversified Functions of Residential Space

By increasing the possibility of converting an interior space to other types of use, it is possible to create maximum conversion potential, prolong the lifespan of the interior space, and increase utilization [45].

Because the area and shape of the room usually limit the function, it is necessary to specify the geometric parameters of the room in order to improve flexibility in residential function transformation. For example, according to the requirements of the DGNB system, the ratio of the usable area of the residence should be between 0.6 and 0.8, the location and area of the traffic space should be reasonable, and the building depth that allows conversion to other uses should be between 11.5m and 13.5m. The living space in each dwelling unit consists of nondedicated rooms, e.g., 3 m × 3 m rooms, ideally 4 m × 4 m [19]. The kitchen and bathroom are set up in a centralized manner, and the connection between the two can be arranged flexibly.

Regarding the flexibility of the house structure, most internal partition walls should be nonload-bearing walls, and the room layout can be changed according to the needs.

4.2.2. Residential Equipment Adjustable

In addition to the actual conditions of the residential building, user satisfaction also depends on the ability to adjust ventilation, sun and glare protection, temperature, and lighting to their liking [46]. Therefore, allowing the occupants to adjust the parameters of each device in the house by themselves, thereby adjusting the indoor physical environment, can improve the user's satisfaction with the indoor environment. The higher the degree of integration of the control system, the more convenient and efficient the user's control of the physical environment of the house is.

Most of the control systems of traditional houses exist independently. Residents need to perform different operations in different places when controlling different physical environment items, such as manually opening windows, pulling curtains, and turning on lights. When introducing an intelligent home control system in residence, it can achieve the integration of maximum control over the physical environment. The intelligent home

control system uses IoT technology to connect the control command information issued by the user's control equipment with the home equipment so that the user can control the house more intensively [47]. Users can realize remote control and use timing functions of electric doors and windows, electric curtains, central air conditioning, and intelligent lighting systems through integrated control panels or mobile terminals, such as computers and mobile phones [48].

### *4.3. Optimize Indoor Environmental Quality*

#### 4.3.1. Maintain Indoor Thermal Comfort

The interior should ensure thermal comfort throughout winter and summer, be suitable for the intended use of the dwelling, and provide appropriate comfort for the occupants [49]. The community should provide residents with suitable indoor temperature, humidity, and ventilation according to different regions' natural environments and climates.

#### 4.3.2. Improve Indoor Air Quality

Indoor air pollution significantly contributes to the personal health risk from inhalation. Harmful substances can directly enter the alveoli through breathing, react with intracellular substances, and cause various adverse effects on human health. The way and time of heating and ventilation affect indoor air quality [50].

Improving indoor air quality is beneficial to the health and well-being of users. There are three primary ways to improve residential indoor air quality: controlling indoor pollution sources, improving air exchange rate, and purification [51].

Regarding indoor pollution source control, indoor materials with volatile organic compounds can be reduced or eliminated, mainly paying attention to the use of plates, coatings, paints, and adhesives. For harmful gases generated daily, local ventilation methods, such as range hoods and exhaust fans in bathrooms and kitchens, can be used. In terms of improving the air exchange rate, it is necessary to reasonably arrange the location and size of the residential vents and promote indoor air circulation through natural methods, such as heat and wind pressure, as much as possible to achieve the purpose of ventilation. Mechanical ventilation can also be used to improve indoor ventilation efficiency. When the outdoor air quality is poor, the residential fresh air system can also introduce fresh and clean air into the room. Air purification improves air quality through filtration, adsorption, photocatalysis, plasma purification, ozone oxidation, and ultraviolet sterilization.

#### 4.3.3. Create a Visually Comfortable Environment

An adequate and uninterrupted supply of daylight and artificial light should be ensured in all indoor areas under continuous use. Visual comfort is the basis for general well-being and productive work. Natural light has a positive effect on human physical and mental health [52]. In addition, the rational use of daylight can save energy required for artificial lighting and maintaining indoor temperature.

Firstly, it should be ensured that there is sufficient natural lighting in the interior of the house. Natural lighting can help regulate people's circadian rhythms, reduce fatigue, improve creativity, and benefit physical and mental health [53]. Visual contact with the outside can also reduce the sense of closure and oppression in the interior. When designing natural lighting, the location, orientation, and lighting angle of the house should be considered. The window-to-wall ratio, window sill height, and window shape of the house's outer wall should be compared to choose the best combination to ensure indoor natural lighting [54]. Secondly, it is necessary to prevent glare from interfering with the interior and indoor heat gain caused by excessive sunlight. Finally, the illuminance, uniformity, stability, and light color of indoor artificial lighting are also crucial in achieving visual comfort in residential buildings [55]. It is also necessary to adopt different light source types and lamp layouts according to different users, usage scenarios, room function positionings, space layouts, indoor materials, and natural lighting environments.

### 4.3.4. Noise Isolation

This objective is to ensure sound insulation is suitable for the room and to prevent unreasonable noise disturbance from the outside.

Adding sound-insulating cushions on concrete floors, using wooden floors and sound-insulating ceilings, increasing the airtightness of residential doors and windows, and filling sound-insulating materials around pipe wells can improve the residential acoustic environment and play a better role in sound insulation.

Protection from disruptive noise has a significant impact on occupant health and satisfaction [56]. Good noise insulation enables users to concentrate better, helps ensure their privacy, provides them with better peace, and improves their health and comfort.

### *4.4. Promote a Healthy Lifestyle*

### 4.4.1. High-Quality Public Space

Studies have shown that social support and social activities can reduce cardiovascular, nervous, and immune system-related morbidity and mortality. Positive social relationships are conducive to maintaining physical and mental health [57]. Therefore, the community should provide users with high-quality indoor and outdoor public spaces to accommodate various recreational and functional uses as much as possible and improve the sustainability of the residence and the comfort of all users over a more extended period.

In the indoor public area of the residence, some areas for communication should be set up, such as seats, lounges, and multifunctional rooms in open places. Public spaces for different purposes, such as living, working, teaching, and learning spaces, should be provided to different users to meet the indoor activity needs of residents, staff, visitors, and other groups. An area can also integrate multiple functions. For example, lounges for work and study can integrate services, such as rest, meals, and networking. The elderly's activity room can provide technical assistance for daily life, online teaching, and teaching training [58]. It is recommended to use glass partitions for public rooms or use windows and glass doors to allow visual communication between the inside and outside of the room and promote social interaction and communication among users.

In order to provide residents with more thoughtful services, some additional services can be provided in public areas, such as a cafeteria, gym, library, laundry, spa, and sauna, which the operating company manages [40].

Designs suitable for families, children, and the elderly should also be considered in public spaces, such as setting up separate mother–infant rooms, children's play areas, and entertainment areas for the elderly in public areas as well as designated family parking spaces in the parking lot, the size of which is suitable for family members, oversized items, and baby buggies.

The entrances, exits, and circulation areas of the residence should be as open as possible and provide enough open space in the vertical direction to allow communication between people on different floors [19]. The traffic space should ensure sufficient natural lighting and ventilation. Some decorative paintings or works of art can also be placed in the traffic space to reduce dullness.

Outdoor courtyards can be used for socializing, and the design should consider materials, lighting, navigation, greenery, and other installations [59]. Outdoor areas should have continuous connections and transitions to the interior to create social spaces and foster a sense of community [19]. Some children's playgrounds, green areas, and parks can be placed outside. Other auxiliary facilities, such as waste disposal sites, bicycle storage facilities, and ventilation in underground garages, should also be integrated into the design of outdoor spaces. At the same time, the courtyard should be equipped with fixed devices and equipment for residents to rest, such as seats, outdoor dining tables, outdoor charging devices, fixed fitness equipment, and shelters.

### 4.4.2. Support Various Modes of Transportation

Traffic exists outside as well as inside the community. People need to become fully prepared inside the community before and after travel. Communities support diverse modes of travel that conserve natural resources and reduce traffic-related air pollution and other negative impacts [60]. Residents should be encouraged to use electric bicycles, electric vehicles, and public transportation [61]. So, they should be provided with public transportation rental systems or mobile platforms to meet personal travel needs.

Green transportation needs the support of indoor facilities. To support the use of bicycles, clearly demarcated areas within or around dwellings should be allocated to bicycle parking facilities [62]. These areas should have high accessibility and support antitheft measures for bicycles, antivandalism measures for parking, and bicycle maintenance facilities. If it is an outdoor parking space, it should also be equipped with appropriate weather protection and lighting. Residents using electric bicycles should be equipped with charging stations or other charging facilities.

In order to support the use of electric vehicles, it is necessary to delineate a particular area in the motor vehicle parking lot and equip charging piles. The charging piles of electric vehicles can be integrated into the residential energy management system for unified management and billing. Other auxiliary indoor facilities can also be equipped near the transportation system, such as shower facilities, changing rooms and drying rooms, storage facilities, spare wheelchairs, strollers, and other facilities to provide residents with a more comfortable and convenient transportation experience.

### 4.5. Build Operation Management System

### 4.5.1. Community Service Facilities

The community will provide community services and related facilities for the residents. At the same time, community boundaries and service facilities can be opened to the public to ensure that the community can better integrate into its urban environment [19].

The community should include offices and service places for community managers, such as property offices, community activity centers, and community service centers. Other functions that provide convenience and assistance to residents' lives can also be added, such as community gyms, medical care, libraries, and daycare centers [63].

### 4.5.2. Community System Check

After the community is built, it needs to be able to be handed over to the owner quickly and to ensure that its systems are running and that all its functions and attributes work as initially designed [19]. Therefore, before the community is put into use, the community management personnel should check the various facilities in the community to ensure the promised construction quality. If necessary, it can be modified according to the actual usage. Before the inspection, a professional inspection team should establish the goals and tasks, inspection scope, inspection basis parameters, and time.

### 4.5.3. Keep Users Informed

The operator of the community should actively inform the residents of the healthy and sustainable use of the community to stimulate and encourage their healthy behaviors, make the community more sustainable and healthier, and ultimately benefit the residents' health [64].

The community operator should provide users with a "healthy development guide", which provides specific action plans for community users on ecological, economic, and social issues, including information on energy conservation, water conservation, garbage classification, and healthy indoor air. The user can choose the form of the guide, such as a paper version, digital version, and regular subscription. Furthermore, all resident users of the community can equally access the guide information updated subsequently. The guidance should also include other information not directly related to the community but relevant to residents, such as the safety of residents, measures to safeguard the com-

munity, recommendations for residents to maintain their health, and suggested modes of transportation near the community [19].

In order to better deliver information about the community to residents, relevant information systems should be installed, thereby incorporating users into the management process of a healthy community by communicating information to residents in real time and obtaining user feedback. The community information system can provide residents with information about the health projects adopted and achieved by the community and future goals through various information media, such as electronic screens, posters, and bulletin boards. An Internet platform can also be established to convey information to residents and provide offline services through virtualized and intelligent online methods [65].

Residents should be regularly informed of the community's management methods and future management plans, and every resident should be informed of community-held activities or convenient services [66]. Community managers should establish smooth information feedback channels to receive residents' difficulties and suggestions to the community in a timely fashion.

## 5. Conclusions

This study explores a series of the DGNB system criteria related to the healthy community and their relationship with community health's three dimensions. The study found that the materials, indoor thermal environment, air quality, and lighting evaluated by DGNB would affect residents' physical health; landscape view, lifestyle, and community safety were associated with mental health; links between DGNB's sociocultural and economic criteria with the community's social health were also evident. Residents' understanding of their own health also focused on personal safety, lifestyle, environment, community service, and management. This study is not only specifying healthy outcomes but also proposing how to achieve them through design, construction, and management. It can be a reference and help for future community designers and developers.

Although the research has achieved preliminary results but is limited by the research time and conditions, there are still many contents to be further studied. In addition to the existing criteria in the DGNB, other health-related design factors that may have an impact should also be investigated. Although the assessment model has identified different indicators, the degree of influence and weight of each indicator should be further studied in future research.

**Author Contributions:** Conceptualization, M.H. and Y.T.; methodology, M.H.; software, M.H. and Y.T.; validation, M.H. and S.Q.; formal analysis, M.H. and S.Q.; investigation, M.H. and Y.C.; resources, Y.T.; data curation, M.H. and Y.C.; writing—original draft preparation, M.H.; writing—review and editing, Y.T.; visualization, M.H. and S.Q.; supervision, Y.T. and Y.C.; project administration, Y.T. and S.Q. All authors have read and agreed to the published version of the manuscript.

**Funding:** This research received no external funding.

**Institutional Review Board Statement:** Not applicable.

**Informed Consent Statement:** Not applicable.

**Data Availability Statement:** Not applicable.

**Acknowledgments:** The authors would like to thank the anonymous reviewers and the editor for their insightful comments and suggestions.

**Conflicts of Interest:** The authors declare no conflict of interest.

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
