# Peer review of "Healthy Community Assessment Model Based on the German DGNB System"

_sustainability, doi:10.3390/su15043167_

Round 1

Reviewer 1 Report

`I enjoy reading the work. However, i have come with following issues..

(1) What is sampling method and i do not see that how 487 responses were obtained.

(2) I do not see any inferential statistics to draw a conclusion

(3) Only Weighted average work is done

(4) there is substantial potential to include inferential statistics and draw conclusions based on that..

Author Response

According to your comments, we have made extensive modifications to our manuscript and supplemented extra data to make our results convincing. Thank you again for your positive comments and valuable suggestions to improve the quality of our manuscript.

Reviewer 2 Report

The article may be accepted by major revision 

Author Response

On behalf of all the contributing authors, I would like to express our sincere appreciation for your constructive comments concerning our article. These comments are all valuable and helpful for improving our article. According to the associate editor and your comments, we have made extensive modifications to our manuscript and supplemented extra data to make our results convincing. In this revised version, our manuscript's changes were highlighted within the document using red-colored text. Point-by-point responses are listed below this letter.

Reviewer 3 Report

In this study authors aimed to establish an evaluation system to evaluate the health of the community, including physical health, social health, mental health and other aspects. The authors provided a comparative analysis between several evaluation systems. I read this work with interest. The manuscript is relatively well written and address a current topic such as the impact of living environment on people's physical and mental health.

REVIEW COMMENTS

It is recommended for publication in Sustainability after the following minor revisions to improve its quality before publication:

1.      Undoubtedly, the topic of the investigation is very relevant and important, but the issue on how residential environments can help to reduce the spread of epidemics is very interesting and needs to be improved in introduction and  in conclusion.

2.      Line 325-326. "......prevent dangerous situations in and around homes, and prevent dangerous situations in and around homes as 326 much as possible.”.   Please rephrase this sentence.

Author Response

We feel great thanks for your professional review work on our article. As you are concerned, there are several problems that need to be addressed. According to your nice suggestions, we have made extensive corrections to our previous manuscript, the detailed corrections are listed below.

Round 2

Reviewer 2 Report

The authors have incorporated all comments. However conclusion still needs some modifications according to results.